# Psychological distress and burnout among healthcare worker during COVID-19 pandemic in India—A cross-sectional study

**Geetha R. Menon** [1]*, **Jeetendra Yadav** [1], **Sumit Aggarwal** [2]*, **Ravinder Singh**[2], **Simran Kaur**[1], **Tapas Chakma**[3], **Murugesan Periyasamy**[4], **Chitra Venkateswaran** [5], **Prashant Kumar Singh**[6], **Rakesh Balachandar**[7], **Ragini Kulkarni**[8], **Ashoo Grover**[9], **Bijaya Kumar Mishra**[10], **Maribon Viray**[11], **Kangjam Rekha Devi**[12], **K. H. Jitenkumar Singh**[1], **K. B. Saha**[3], **P. V. Barde**[3], **Beena Thomas**[4], **Chandra Suresh**[4], **Dhanalakshmi A.**[4], **Basilea Watson**[4], **Pradeep Selvaraj**[13], **Gladston Xavier**[14], **Denny John**[15], **Jaideep Menon**[15], **Sairu Philip**[5], **Geethu Mathew**[5], **Alice David**[5], **Raman Swathy Vaman** [5], **Abey Sushan** [5], **Shalini Singh**[6], **Kiran Jakhar**[16], **Asha Ketharam**[7], **Ranjan Prusty** [8], **Jugal Kishore**[17], **U. Venkatesh** [17], **Subrata Kumar**[10], **Srikanta Kanungo**[10], **Krushna Sahoo**[10], **Swagatika Swain**[10], **Anniesha Lyngdoh**[11], **Jochanan Diengdoh**[11], **Phibawan Syiemlieh**[11], **AbuHasan Sarkar**[12], **Gajanan Velhal**[18], **Swapnil Kharnare**[19], **Deepika Nandanwar**[18], **M. Vishnu Vardhana Rao**[1], **Samiran Panda**[2]

1 Indian Council of Medical Research- National Institute of Medical Statistics, New Delhi, India, 2 Indian Council of Medical Research, New Delhi, India, 3 Indian Council of Medical Research- National Institute of Research in Tribal Health, Jabalpur, Madhya Pradesh, India, 4 Indian Council of Medical Research- National Institute of Research in Tuberculosis, Chennai, Tamil Nadu, India, 5 Believers Church Medical College Hospital, Tiruvalla, Kerala, India, 6 ICMR-National Institute of Cancer Prevention and Research, Gautam Buddh Nagar, Uttar Pradesh, India, 7 ICMR-National Institute of Occupational Health, Ahmedabad, Gujarat, India, 8 ICMR- National Institute of Research in Reproductive Health, Mumbai, Maharashtra, India, 9 ICMR- National Institute of Pathology, New Delhi, India, 10 Regional Medical Research Centre, Bhubaneswar, Odisha, India, 11 Martin Luther Christian University, Shillong, Meghalaya, India, 12 ICMR -Regional Medical Research Centre, Dibrugarh, Assam, India, 13 Directorate of public health and Preventive Medicine, Chennai, Tamil Nadu, India, 14 Loyola College, Chennai, Tamil Nadu, India, 15 Amrita Institute of Medical Sciences & Research Centre, Kochi, Kerala, India, 16 Government Institute of Medical Sciences, Greater Noida, Uttar Pradesh, India, 17 Vardhman Mahavir Medical College and Safdarjung Hospital, New Delhi, India, 18 Seth GS Medical College and KEM Hospital, Mumbai, Maharashtra, India, 19 Hinduja Hospital, Mumbai, Maharashtra, India

* menongr.hq@icmr.gov.in (GRM); drsumiticmr@gmail.com (SA)

**Data Availability Statement:** All relevant data are within the paper and its Supporting Information files.

## Abstract

### Background

COVID-19 has inundated the entire world disrupting the lives of millions of people. The pandemic has stressed the healthcare system of India impacting the psychological status and functioning of health care workers. The aim of this study is to determine the burnout levels and factors associated with the risk of psychological distress among healthcare workers (HCW) engaged in the management of COVID 19 in India.

### Methods

A cross-sectional study was conducted from 1 September 2020 to 30 November 2020 by telephonic interviews using a web-based Google form. Health facilities and community centres from 12 cities located in 10 states were selected for data collection. Data on socio-

**Funding:** The study was conducted under the aegis of Indian Council of Medical Research which has also funded the study.

**Competing interests:** The authors have declared that no competing interests exist.

**Abbreviations:** HCW, Healthcare Workers; GHQ-5, General Health Questionnaire-5; ICMR, Indian Council of Medical Research; NIOH, National Institute of Occupational Health; AOR, Adjusted Odds Ratio; UOR, Unadjusted Odds Ratios; OR, Odds Ratio; CI, Confidence Intervals; PPE, Personal Protective Equipment; ICD, International Classification of Diseases; MBI, Maslach Burnout Inventory; EE, Emotional Exhaustion; DP, Depersonalisation; PA, Personal Accomplishment; PTSD, Posttraumatic Stress Disorder; SARS, Severe Acute Respiratory Syndrome; MERS, Middle East Respiratory Syndrome; HAM-A, Hamilton Anxiety Rating Scale; IEC, Institutional Ethics Committees; CECHR, Central Ethics Committee on Human Research.

demographic and occupation-related variables like age, sex, type of family, income, type of occupation, hours of work and income were obtained was obtained from 967 participants, including doctors, nurses, ambulance drivers, emergency response teams, lab personnel, and others directly involved in COVID 19 patient care. Levels of psychological distress was assessed by the General health Questionnaire -GHQ-5 and levels of burnout was assessed using the ICMR-NIOH Burnout questionnaire. Multivariable logistic regression analysis was performed to identify factors associated with the risk of psychological distress. The third quartile values of the three subscales of burnout viz EE, DP and PA were used to identify burnout profiles of the healthcare workers.

## Results

Overall, 52.9% of the participants had the risk of psychological distress that needed further evaluation. Risk of psychological distress was significantly associated with longer hours of work ($\geq$ 8 hours a day) (AOR = 2.38, 95% CI(1.66–3.41), income$\geq$20000(AOR = 1.74, 95% CI, (1.16–2.6); screening of COVID-19 patients (AOR = 1.63 95% CI (1.09–2.46), contact tracing (AOR = 2.05, 95% CI (1.1–3.81), High Emotional exhaustion score (EE $\geq$16) (AOR = 4.41 95% CI (3.14–6.28) and High Depersonalisation score (DP$\geq$7) (AOR = 1.79, 95% CI (1.28–2.51)). About 4.7% of the HCWs were overextended (EE>18); 6.5% were disengaged (DP>8) and 9.7% HCWs were showing signs of burnout (high on all three dimensions).

## Conclusion

The study has identified key factors that could have been likely triggers for psychological distress among healthcare workers who were engaged in management of COVID cases in India. The study also demonstrates the use of GHQ-5 and ICMR-NIOH Burnout questionnaire as important tools to identify persons at risk of psychological distress and occurrence of burnout symptoms respectively. The findings provide useful guide to planning interventions to mitigate mental health problems among HCW in future epidemic/pandemic scenarios in the country.

## Introduction

The Coronavirus disease 2019 (COVID-19) was first identified in Wuhan in China in December 2019 [1] and has now spread to 220 countries leading to 194.1 million confirmed cases and 4.2 million deaths [2]. As on July 26,2021 India had reported 31.02 million cases and 0.42 million deaths from the day the first case was seen on January 30, 2020 [3]. Worldwide the pandemic has impacted the physical and mental health of the frontline health workers than the general population. During the initial phase of the pandemic, the health care workers (HCW) faced plenty of challenges because of the novel nature of the disease, limited treatment options, fear of infection of self and their loved ones, shortages of personal protective equipments (PPE), extended workloads, and facing difficulties in making emotionally and ethically difficult triaging and resource-allocation decisions [4]. A number of health care workers have shown hesitancy to go to work thereby facing the loss of jobs and reduced revenues. The unknown nature of the disease and also the conflicting alternatives of treatment and management have tested the tolerance of the patient's relatives. In some parts of India, doctors and health care

workers have faced stigma, abuse, and violence [5]. These conditions have had a significant impact on the psychological state of frontline health workers that was an imminent concern.

The overall psychological distress has been linked to burnout, which could be work-related professional hazard acquired when providing healthcare for patients. Psychological distress is a state of emotional suffering resulting from being exposed to a stressful situation that poses a threat to one's physical and mental health [6]. Psychological distress can manifest into adverse mental state and psychiatric outcomes including depression, anxiety, acute stress, post-traumatic stress and burnout. These may negatively impact day to day and social functioning of an individual [7]. Burn-out is a psychological term for a negative response to chronic workplace stress. It is said to occur when people give an excessive amount of their time, energy and efforts on their job over an extended period of time without enough time to recover physically or emotionally [8]. In ICD-11, burnout is a syndrome conceptualized as resulting from chronic workplace stress that has not been successfully managed [9]. It is characterized by three dimensions: emotional exhaustion associated with feelings of energy depletion or exhaustion; depersonalisation refers to disengagement from work due to over exhaustion and personal accomplishment that refers to feelings of competence, achievement and accomplishment in one's work. The present study was undertaken 1) to assess the occurrence of psychological distress and burnout among healthcare workers from different cities in India and 2) to explore the factors related to the risk of psychological distress and burnout by considering both personal and work-related characteristics.

## Methods

### Study design

This is a multi-centric Indian Council of Medical Research (ICMR) Task Force study to assess the occurrence of psychological distress in health care workers engaged in the management of COVID-19 patients and to gain insight on their burnout levels due to the impact of COVID. It was conducted during 1$^{st}$ September 2020 and 30th November 2020. The study used a cross sectional design assuming 50% prevalence of psychological distress, with an alpha error of 5% and relative precision of 10%. A sample size of 385 was estimated using the formula $n = \frac{1.96^2}{l^2} p(1 - p)$. Assuming 15% noncompliance rate, the sample size was inflated to 452. Accounting for a design effect of 2.0 for possible clustering in view of including health personnel from the same facility, the minimum sample size required was 452X2 = 904 at a national level. Actual data was collected from 967 participants.

### Setting and participants

The study was conducted in ten states of the country. The study sites purposively selected across these states included Bhubaneswar and Cuttack (Odisha), Mumbai (Maharashtra), Ahmedabad (Gujarat), Noida (Uttar Pradesh), South Delhi, Pathanamthitta and Kasargod (Kerala), Chennai (Tamil Nadu), Jabalpur (Madhya Pradesh), Kamrup (Assam) and East Khasi Hills (Meghalaya) (Fig 1).

The site investigators contacted the health authorities of each site, explained the purpose of the study, and sought their cooperation to carry out the study. A list public and private hospitals in each site that were involved in COVID-19 management and care services was prepared and those willing to participate, were shortlisted for the conduct of the study. ICMR sent out a letter of appeal for support and cooperation to the chosen facilities before the study. Health care workers involved in triaging, screening, treatment, isolation, referral services and community outreach services related to COVID-19 management were identified. These included

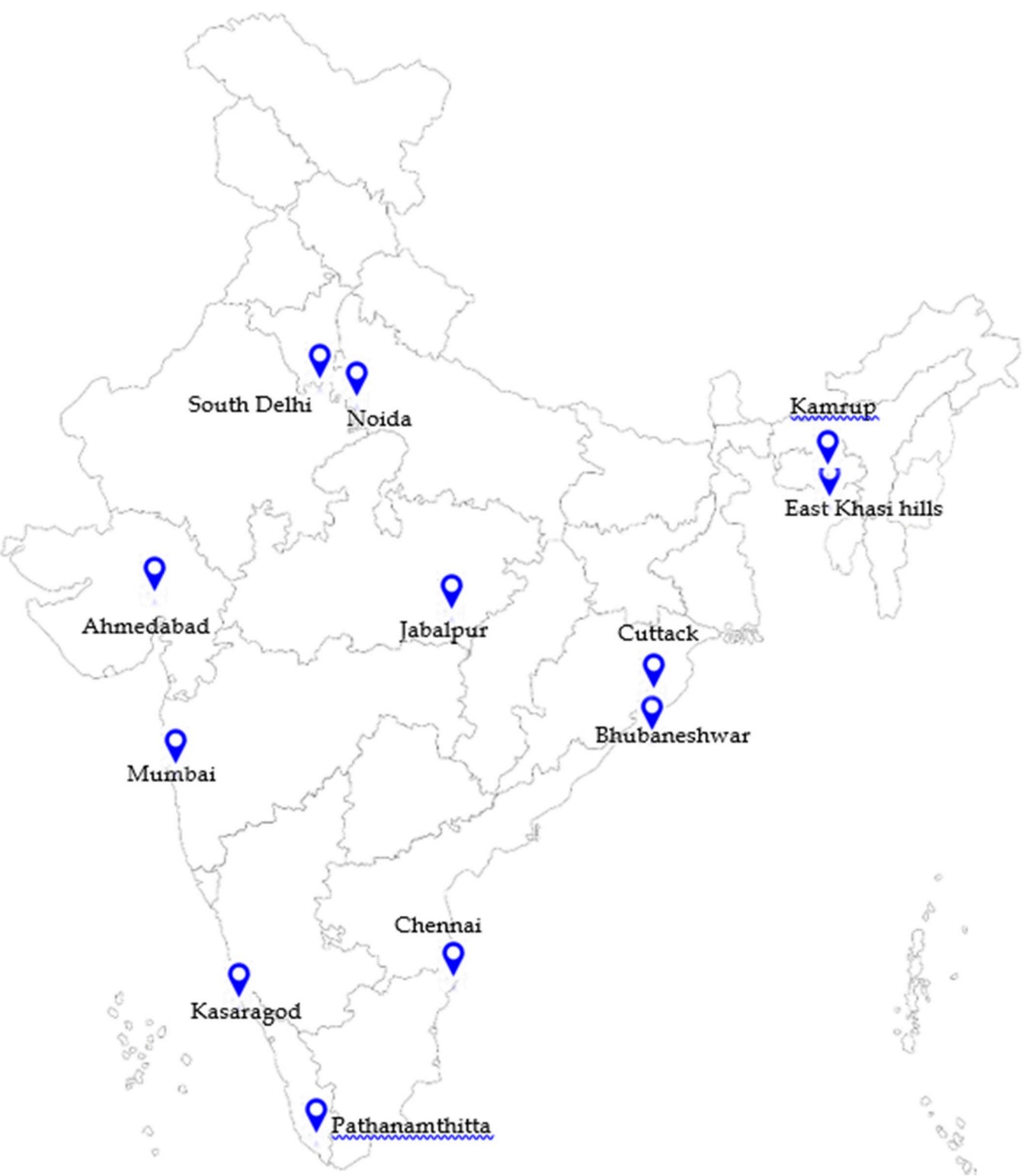

**Fig 1. Map showing study sites.** Source: https://mapchart.net/india.html.

doctors, nurses, pharmacists, ambulance workers, community workers, housekeeping staff, security guards, stretcher-bearers, sanitation workers, laboratory staff and hospital attendants. The investigators individually contacted the eligible participants telephonically, explained the purpose of the study with the help of a participant information sheet, and their willingness to participate was obtained through audio consent. The participant's anonymity and confidentiality of information was ensured. The investigators fixed an appointment with them as per their convenience so as to ensure that their duty time or leisure time was not disturbed. All interviews were conducted telephonically in view of the ongoing pandemic. Each interview lasted for 20–30 minutes. The field investigators filled the printed questionnaire during interview which was scrutinised by the site investigators for completeness before entering the data in the data entry template. Each state was expected to collect data of a minimum of 90 participants to meet the sample size requirement at the national level.

## Data collection & study tool

Data was collected on a semi structured questionnaire (S1 File) which was translated in the local language by the site investigators. The questionnaire was made up of three parts. Part I contained information on the facility (public or private) a respondent was attached with, personal background, and job-related factors like average number of COVID patients screened per day and the average number of COVID patients under care per day etc. Detailed information was collected on age, sex, marital status, education, number of family members, residing with family, monthly income, place of work(institutional or community), average number of working hours, type of occupation, workplace characteristics (institutional or non-institutional) hours of work per day, type of activities involved in COVID-19 care (quarantine, isolation, intensive care, bereavement, contact tracing, community care, screening and transport) and hydroxychloroquine prophylaxis. The patient information form and consent form were included in Part I (S1 File).

**Assessment of risk of psychological distress.** Part II of the questionnaire was the General Health Questionnaire-5 (GHQ-5). The 12-item General Health questionnaire is a well-validated indicator of psychological distress [10]. The advantage of GHQ-12 is that it is short, can be easily scored "clinically" (symptoms present or absent) as well as levels of symptoms present (Likert-type scoring). The instrument is frequently used in screening of civilian populations in different cultures. The GHQ-5, a shorter screening tool which has 5 questions with better discriminators for psychological distress derived from GHQ-12, is validated on the Indian population with a sensitivity of 86%, specificity of 95.8% and misclassification rate of 8.3% [11]. For this study that was done telephonically and needed quick assessment of the psychological state of the respondent with minimum number of questions, GHQ-5 was an effective tool with regard to time and process. It is also available for use in public domain as compared to GHQ-12 that involves royalty for usage. GHQ-5 has five questions viz. have you recently lost much sleep over worry, have you recently felt constantly under strain, have you recently been able to enjoy normal day to day activities, have you recently been feeling reasonably happy all things considered, have you recently been feeling unhappy and depressed? The respondent is expected to respond with either "Yes" or "No" for each of the questions. "Yes" is scored as 1 and "No" is scored as 0 for Q1, 2 and 5 while the remainder questions (Q3 and Q4) are coded in the reverse i.e. No is scored as 1 and Yes is scored as 0. Individuals with a score of $\geq 2$ are suggestive of psychological distress and warrant further evaluation for psychological /psychiatric morbidity by the mental health expert.

**Burnout assessment.** The ICMR-NIOH Burnout Questionnaire [12] was Part III of the questionnaire. This questionnaire is a shortened and easier version developed by

ICMR-National Institute of Occupational Health in Indian settings of the Maslach Burnout inventory(MBI) [13]. It was freely available for use whereas the MBI inventory is copyrighted and cannot be reproduced without permission from the developers. It is a 19-item questionnaire that covers three dimensions of burnout: a) emotional exhaustion (EE, 11 items), which describes the sense of getting one's emotional resources exhausted and no way to replenish them. EE subscale items describe feelings of being emotionally overextended and exhausted by one's work; (b) depersonalisation (DP, 5 items) describes the experience of becoming cold and indifferent to the need of others and (c) personal accomplishment (PA, 3 items). The five items within the DP subscale measure an unfeeling and impersonal response toward recipients of one's service, care treatment or instruction. Higher mean scores on the EE and DP subscales indicate high degrees of experienced burnout. The three items in the PA subscale assess the feelings of competence and achievement while in workplace Lower PA mean scores correspond to higher degrees of experienced burnout in contrast to EE and DP subscales. The PA subscale is independent of the other subscales, and its component items don't load negatively on them. In other words, PA cannot be assumed to be the opposite of EE or DP. The items in each subscale are answered in terms of the frequency with which the respondent experiences these feelings. For the study, the score was performed on a 3-point scale starting from 1," never," 2, "sometimes" and 3, "always". In order to ensure the attention of the respondents, the questions have an inverse scoring system, e.g. Q10 of emotional exhaustion; the direction of scoring to Q10 is inverse to the rest of the questions. The scores for every subscale are considered separately; thus, three scores are computed for every respondent.

## Statistical analysis

Statistical analysis was performed using IBM Corp. Released 2015. IBM SPSS Statistics for Windows, Version 23.0. Armonk, NY: IBM Corp.). First, univariate analysis was done by summarising the sociodemographic characteristics for low (GHQ<2) and high risk (GHQ> = 2) of psychological distress. Age, sex, marital status, living condition, salary level, educational level, average daily hours of work, place of work, occupation were presented as frequencies and percentages and compared using $\chi^2$ test for categorical variables. The cut-off values of Emotional exhaustion, Depersonalisation and Personal accomplishment was obtained by plotting the mean EE, DP and PA scores for each GHQ score from minimum (0) to maximum (5). The EE, DP and PA cut-offs were summarised using $\chi^2$ test for low and high risk of psychological distress. Univariate analyses between each predictor and outcome (GHQ) were performed and unadjusted odds ratios (UOR) and 95% confidence intervals (CI) were reported. Predictors with univariate p values < 0.20 were included the multivariate analysis.

In multivariate analyses, the relationship of the predictors with GHQ was obtained by estimating the adjusted odds ratios (AOR) and 95% Confidence Intervals (CI). We used the multiple logistic regression, with GHQ score as the binary dependent variable defined as GHQ<2 = 0 and GHQ≥2 = 1 respectively and the variables found significant in univariate analyses were the independent variables. The logistic model for the study was $ln \frac{p}{1-p} = a + b_1 x_1 + b_2 x_2 + b_3 x_3 + \ldots b_n x_n$ where p is probability that GHQ> = 2 i.e at risk of psychological distress, $b_1, b_2 \ldots \ldots b_n$ are the slopes and a is the intercept of the best fitting equation in the multiple logistic regression. The goodness of fit for the logistic model was measured using Nagelkerke $R^2$. The alpha level was assumed to be 0.05 for all effects.

We categorized the burnout scores into five profiles based on the third quartile values of the three subscales EE, DP and PA. Thus, the categories were Burnout: high scores on all three dimension; *Overextended*: high on exhaustion only; *Ineffective*: high on inefficacy only *Disengaged*: high on depersonalization only; *Engagement* low on all three dimensions.

### Ethical approval

Ethical approval for this study was obtained from the ICMR- National Central Ethical Committee and the Institutional Ethics Committees (IEC) of each implementing ICMR Institute and other agencies- Reference Number: CECHR 012/2020.

## Results

A total of 967 health care personnel participated in the study. The sociodemographic and work-related characteristics of the study sample are detailed in Table 1. About 62 percent of the respondents were below 34 years, 54.4% were females and 61.2% were married. Sixty-eight percent had graduate-level education and above and 70% were residing with their family members. About one-third (33.0%) were getting a salary between Rs. 20000–50000 while almost one forth fell in the salary levels of 10000–20000 and above Rs. 50000. About 80% of the respondents resided in urban areas. About one-fifth were nurses, while 17 percent were doctors and 16.8 percent ambulance drivers. About 62 percent were temporary employees, 69.2 percent reported working up to 8 hours in a day and 3 percent worked for more than 12 hours in a day.

To obtain the cut-offs for EE, DP and PA, the mean EE scores, mean DP scores and mean PA scores were plotted against each value of GHQ (Fig 2). As the GHQ score increased from 0–5, the mean EE score and DP score showed an increasing trend from 13.36 to 18.28 and 5.87 to 7.34, respectively, indicating a positive association of burnout with psychological distress. However, the mean PA score did not vary much across the GHQ scores. At the GHQ score of 2 the corresponding mean EE score was assumed to be 16, DP score 7 and PA score was 8. For GHQ $<2$ and $> = 2$, the corresponding cut-offs for EE was low = '$<16$' and high = '$> = 16$'; for DP low = '$<7$' and high = '$> = 7$'; for PA low = '$<8$' and high = '$> = 8$'.

### Sociodemographic, job-related factors and risk of psychological distress

As shown in Fig 3, 52.9% (512/967) of the participants showed high risk of psychological distress (GHQ$> = 2$). These respondents were advised to meet psychiatrists for further evaluation. In the unadjusted logistic regression analysis, a female health worker (UOR = 1.32, 95% CI (1.02–1.70)), those living alone (UOR = 1.34, 95% CI (1.02–1.77)), those with monthly income more than 20000 INR (UOR = 2.30 95% CI (1.73–3.04)), those with a graduate or higher degree (UOR = 1.93, 95% CI (1.47–2.54), those working for more than 8 hours/day (UOR = 2.10 95% CI (1.58–2.79), doctors and nurses (UOR = 2.25, 95% CI (1.72–2.94)), those with a higher Emotional exhaustion score (UOR = 5.8, 95% CI (4.3–7.7) and those with a high Depersonalisation score (UOR = 3.3, 95% CI (2.5–4.3))) were more likely to be in psychological distress Age, marital status and place of work(community or health facility) had no association with risk of psychological distress. After adjusting for these factors in the multiple logistic regression, it was observed that health workers working for more than 8 hours/day were two times more at risk of being psychologically distressed as compared to those working for lesser duration (AOR = 2.38,95%CI, (1.66–3.41)); those with income more than 20000 INR were more likely to be psychologically distressed as compared to the lower-income group(AOR = 1.74, 95% CI, (1.16–2.6)); those with high EE score were our times at risk of psychological distress (AOR = 4.41, 3.14–6.28)), and those with high DP score were two times more likely to be at risk of psychological distress (AOR = 1.79, 95% CI (1.28–2.51)) as compared to those with low DP scores.

### Type of activity and psychological distress

The health workers reported doing more than one activity while managing COVID-19patients. Among the activities, health workers involved in isolation of COVID cases (57.9%), caring the

**Table 1. Socioeconomic and job-related characteristics of participants.**

| Background Characteristics | N = 967 | % |
|---|---|---|
| *Age (in years)* | | |
| *< = 34 years* | 600 | 62.1 |
| *35–44 years* | 226 | 23.4 |
| *> = 45 years* | 141 | 14.6 |
| *Sex* | | |
| *Male* | 441 | 45.6 |
| *Female* | 526 | 54.4 |
| *Marital Status* | | |
| *Married* | 592 | 61.2 |
| *Others* | 375 | 38.8 |
| *Education* | | |
| *Graduate and above* | 661 | 68.4 |
| *Below graduate* | 306 | 31.6 |
| *Living condition* | | |
| *With family/other relatives* | 675 | 69.9 |
| *Alone* | 291 | 30.1 |
| *Family size* | | |
| *Up to 3 members* | 219 | 23.3 |
| *3–4 members* | 477 | 50.6 |
| *Five and above* | 246 | 26.1 |
| *Monthly income (Missing 130)* | | |
| *< 10000* | 120 | 14.3 |
| *10000–20000* | 219 | 26.2 |
| *20000–50000* | 276 | 33.0 |
| *50000 and above* | 222 | 26.5 |
| *Place of residence* | | |
| *Rural* | 156 | 16.1 |
| *Urban* | 768 | 79.4 |
| *Semi Urban* | 43 | 4.5 |
| *Designation* | | |
| *Doctors* | 173 | 17.9 |
| *Auxiliary nurse / paramedical staff* | 103 | 10.7 |
| *Nurses* | 190 | 19.7 |
| *Laboratory staff/ Supporting staff* | 142 | 14.7 |
| *House-keeping /sanitation* | 89 | 9.2 |
| *Ambulance driver/staff/ward boys/Guards* | 162 | 16.8 |
| *ASHA/UHW/USHA* | 108 | 11.2 |
| *Place of work* | | |
| *Community* | 108 | 11.2 |
| *Health facility/hospital* | 859 | 88.8 |
| *Employment Status* | | |
| *Temporary* | 605 | 62.6 |
| *Permanent* | 362 | 37.4 |
| *Types of facility* | | |
| *Private* | 200 | 20.7 |
| *Public* | 767 | 79.3 |
| *Hours of working/day* | | |

*(Continued)*

**Table 1.** (Continued)

| Background Characteristics | N = 967 | % |
|---|---|---|
| *Up to 8 hours* | 669 | 69.2 |
| *8–12 hours* | 269 | 27.8 |
| *More than 12 hours* | 29 | 3.0 |

patients with symptoms (58%), working in intensive care units (58.3%), contact tracing (60.1%) screening (59.9%) and in transporting COVID, showed a significant association with psychological distress. Multivariate logistic regression analysis was conducted with socioeconomic and job-related characteristics and type of activities as independent variables and GHQ score as the binary dependent variable. Health workers involved in screening and contact tracing were more likely to be at risk of psychological distress (AOR = 1.63 95% CI (1.09–2.46) and (AOR = 2.05, 95% CI (1.1–3.81) respectively) when the model was adjusted for sociodemographic and job-related variables. The other significant factors were long working hours

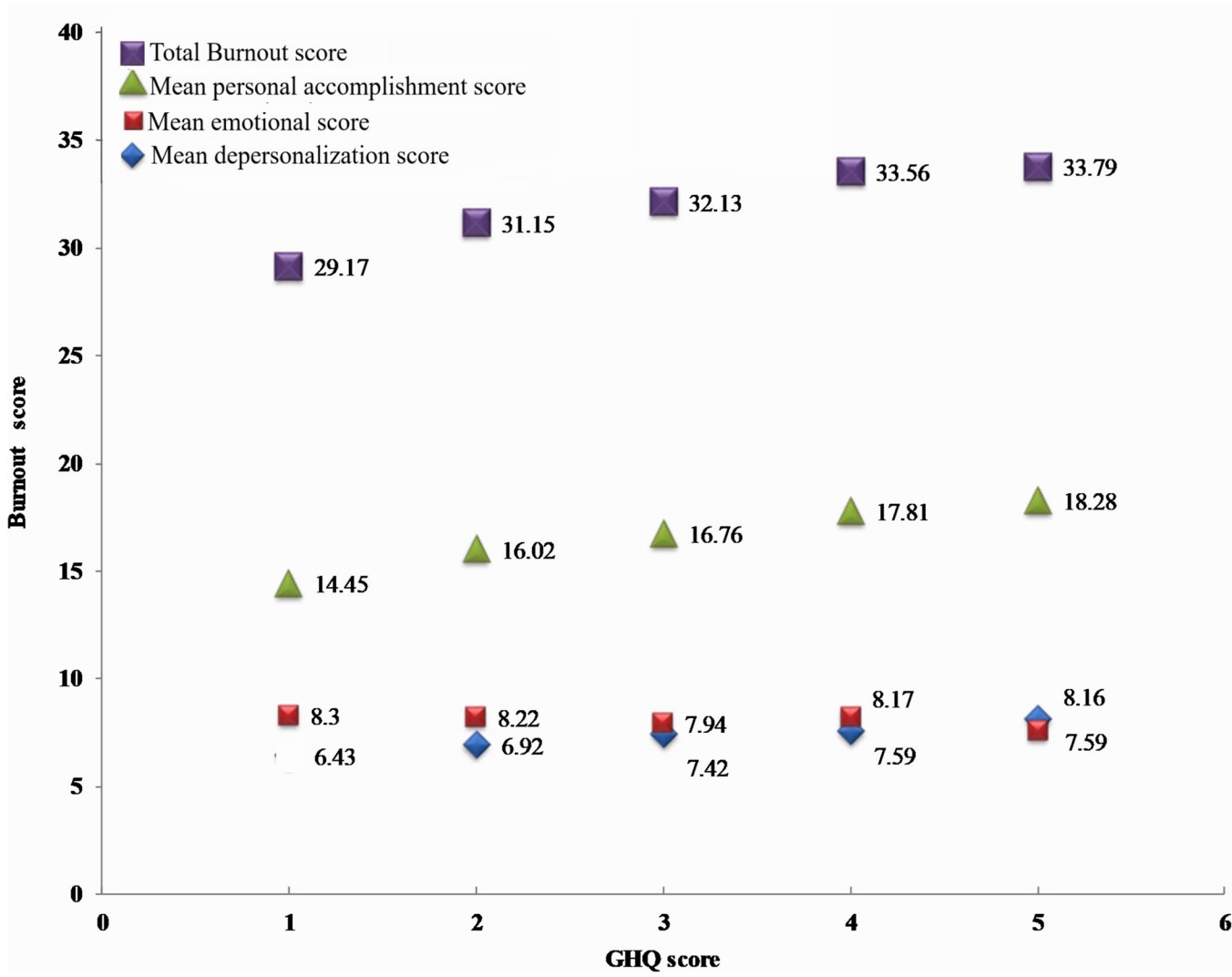

**Fig 2. Mean burnout subscale scores plotted against GHQ scores.**

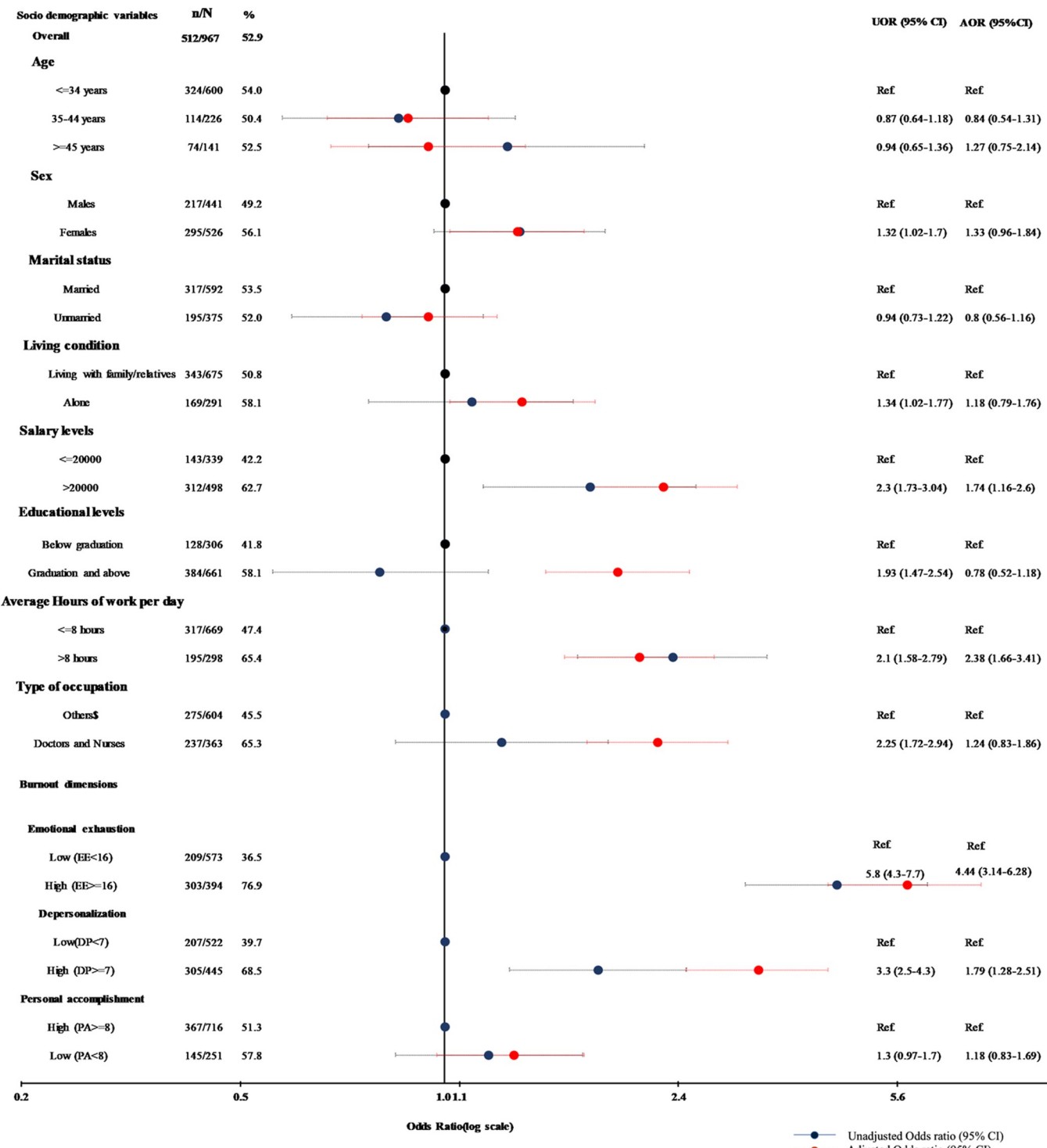

**Fig 3. Odds ratios for the association of socio-economic and job-related variables with the risk of psychological distress.** $^{\$}$Auxiliary nurse / paramedical staff, Laboratory staff/ Supporting staff, House-keeping /sanitation, Ambulance driver/staff/ward boys/Guards, ASHA/UHW/USHA.

(> = 8 hours) (AOR = 2.5, 95% CI (1.7–3.6); higher income (> = 20000INR) (AOR = 1.6, 95% CI (1.1–2.4); High EE score(EE> = 16)(AOR = 4.8, 95% CI(3.3–6.8) and high DP score(DP> = 7) (AOR = 1.9, 95% CI(1.4–2.7) (Fig 4).

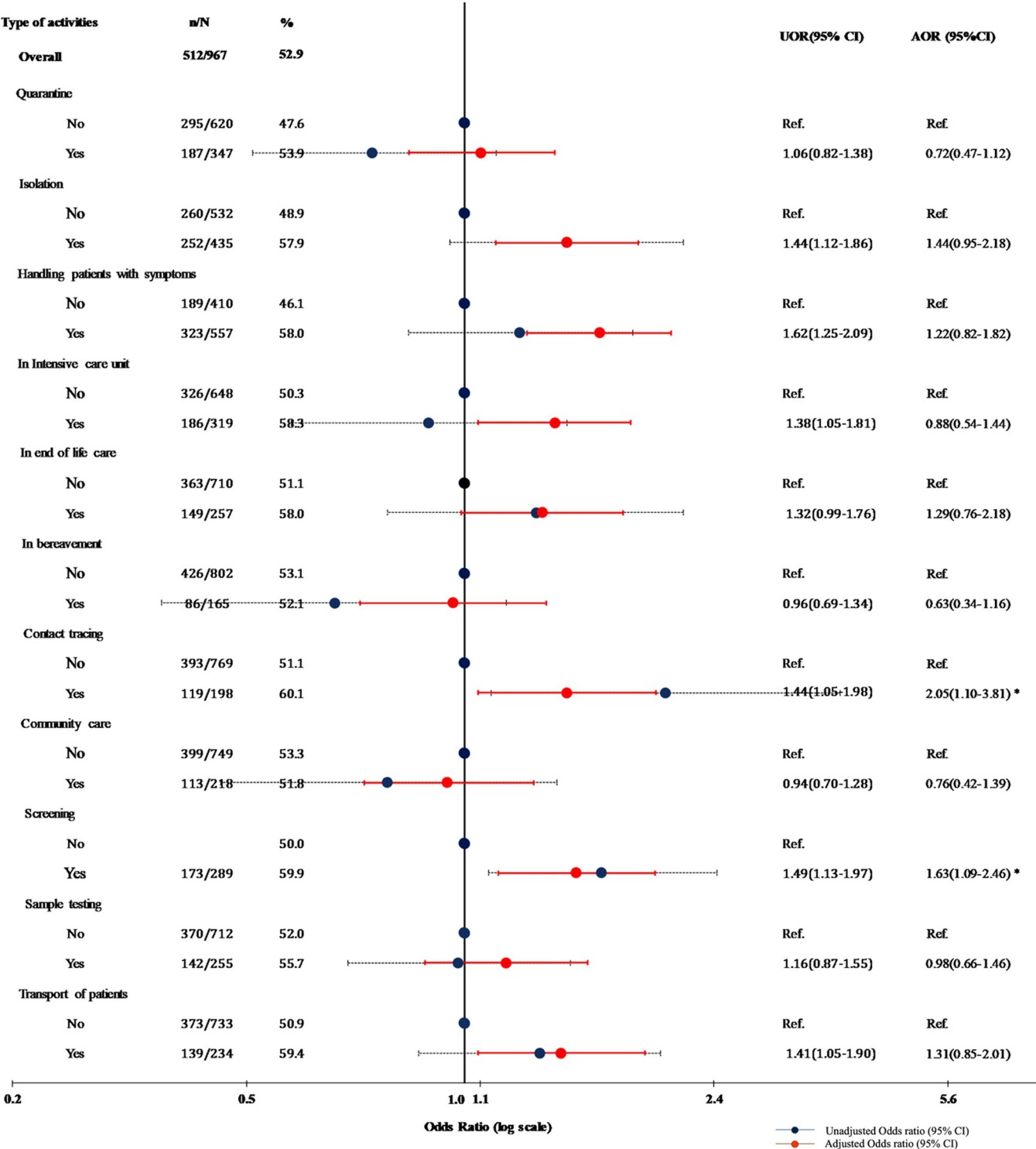

**Fig 4. Odds ratios for the association of types of activities and risk of psychological distress.**

## Sociodemographic, job-related factors and risk of burnout

Within the dimension of emotional exhaustion, nearly 50% of the HCWs reported that they most often kept thinking about work-related issues even during off duty hours, which

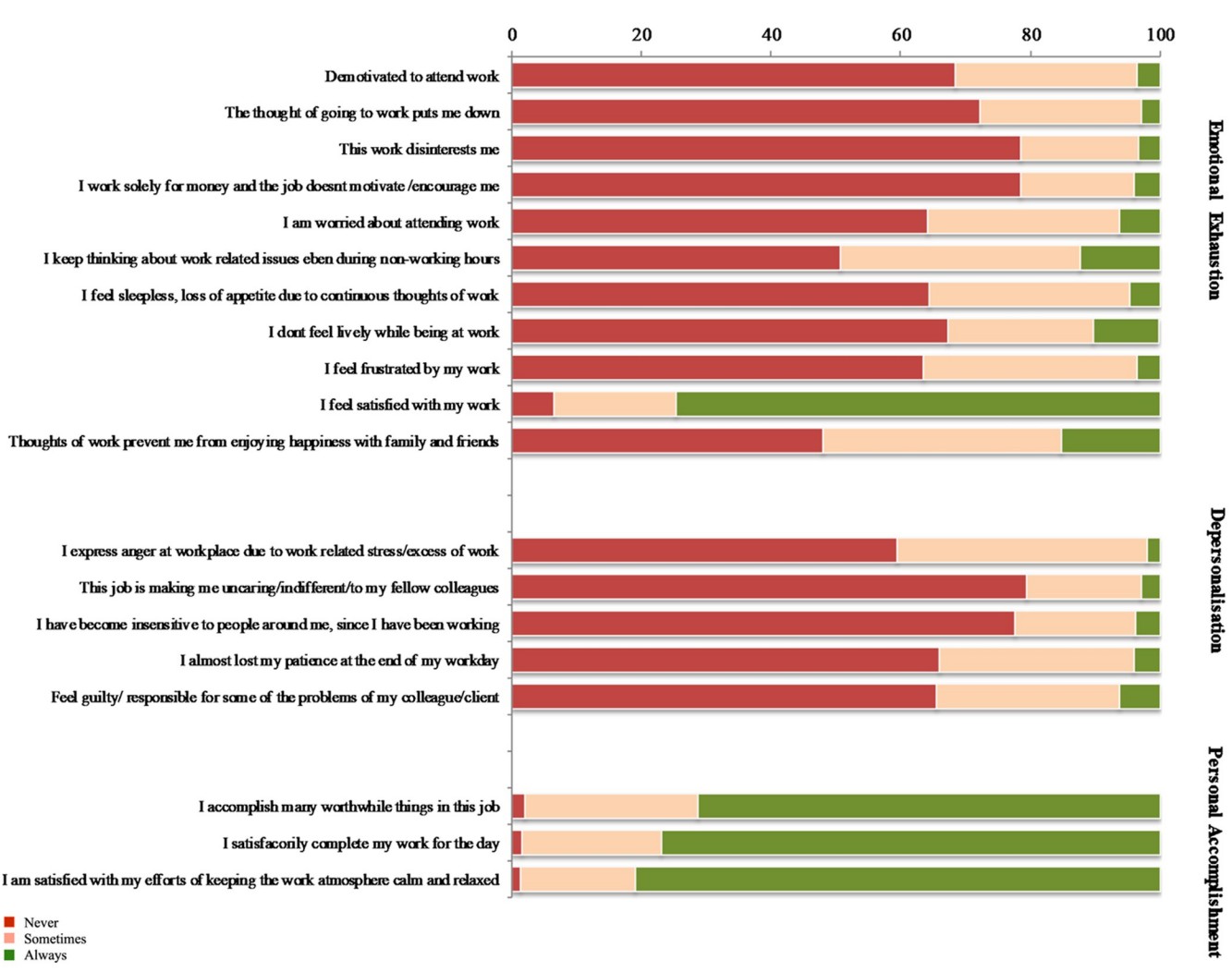

**Fig 5. Distribution of participants in three subscales of burnout.**

prevented them from enjoying with their families. Nearly 35% of the HCWs felt sleepless and had loss of appetite, felt frustrated and constantly worried about their work. However, more than 75% of the HCWs felt satisfied with their work despite all work-related problems. Longer working hours of more than 8 hours per day increased the Emotional exhaustion and depersonalization scores. The job being temporary or permanent had no effect on any subscale of the burnout score. Over 20–40% of the healthcare workers reported a higher score on each item of depersonalization for eg 40% expressed anger at workplace, lost patience at workplace and also felt guilty, 20% responded having an indifferent attitude with fellow colleagues or insensitive to people around (Fig 5).

Since the ICMR-NIOH burnout questionnaire [12] is relatively a naïve tool and validated in a small population it was not possible for us to clearly identify individuals with burnout. However, we applied the concept of Leiter and Maslach [14] who identified five profiles from the MBI viz.

- *Burnout*: high scores on all three dimensions

- *Overextended*: high on exhaustion only

- *Ineffective*: high on inefficacy only

- *Disengaged*: high on depersonalization only

- *Engagement*: low on all three dimensions

To categorise the individuals in these five profiles, we computed the third quartile (Q3) of each subscale score. The Q3 values for EE was 18; for DP 8 and for PA it was 9. The individuals scoring higher than the Q3 values of EE and DP and lower than the Q3 value for PA were categorised as high on these dimensions. With this classification about 4.7% (45/967) of the HCWs were overextended; 6.5% (63/967) disengaged; and 9.7% (94/967) of the HCWs were showing signs of burnout. However, 42% (406/967) of the HCWs were engaged, 5.3% (51/967) were either overextended or disengaged (About 12% were either overextended or disengaged and also ineffective (Fig 6).

The mean and median EE scores, DP scores and PA scores across different age groups, income levels, occupation, type of living arrangement and hours of work were compared with the three subscales of burnout. The median EE score was statistically significant in respondents living alone as compared to those living with family (15 vs. 14, p = 0.03), those who were working for more than 8 hours as compared to less than 8 hours per day (15 vs. 14, p = 0.008), those who were graduate and above, those with income >20000(15 vs. 14, p = 0.001) and for doctors and nurses (16 vs. 14, p = 0.001). The median DP scores varied significantly across all demographic variables except gender and employment status. Respondents below the age of 35 years (7 vs. 6 p = 0.001), who were single (7 vs. 6, p = 0.001), doctors and nurses (7 vs. 6, p = .001), with higher income levels (7 vs. 6, p = 0.001) had significantly higher mean /median DP scores (S1 Table).

## Discussion

Overall, we found that more than half of the frontline health workers, who provided intensive care, who were involved in tracing and screening and transporting patients were at risk of psychological distress and needed further psychiatric evaluation. Similar studies have been conducted in China, Italy and Singapore on health care personnel during the outbreak of the COVID-19 pandemic [15–19]. These studies have shown that health care personnel were more likely to suffer from symptoms of depression, anxiety, and posttraumatic stress disorder (PTSD).

Previous studies from SARS, MERS or Ebola outbreaks have shown that the onset of sudden illnesses has resulted in psychological distress and posttraumatic stress among the health care workers [19–21]. During the past epidemics and crisis, studies in many parts of the world including India have reported anxiety and fear [20–23]. A rapid review of studies on mental health of HCWs found a high proportion of individuals with depression, anxiety, stress, posttraumatic stress, insomnia and burnout [24]. Kisley et al. [25] stated that such outbreaks resulted in psychological distress and posttraumatic stress among the HCWs that had many determinants like close contact with affected patients, forced redeployment to manage infected patients, inadequate training to use PPE, fear of quarantine from family and societal factors (societal stigma against hospital workers) which are also important in an Indian healthcare scenario. Studies done after the SARS outbreak in 2003 have shown that healthcare professionals who were working in high-risk environments showed adverse psychological outcomes, leading to work performance decline [26]. Another Chinese study, Lai et al., revealed that HCWs

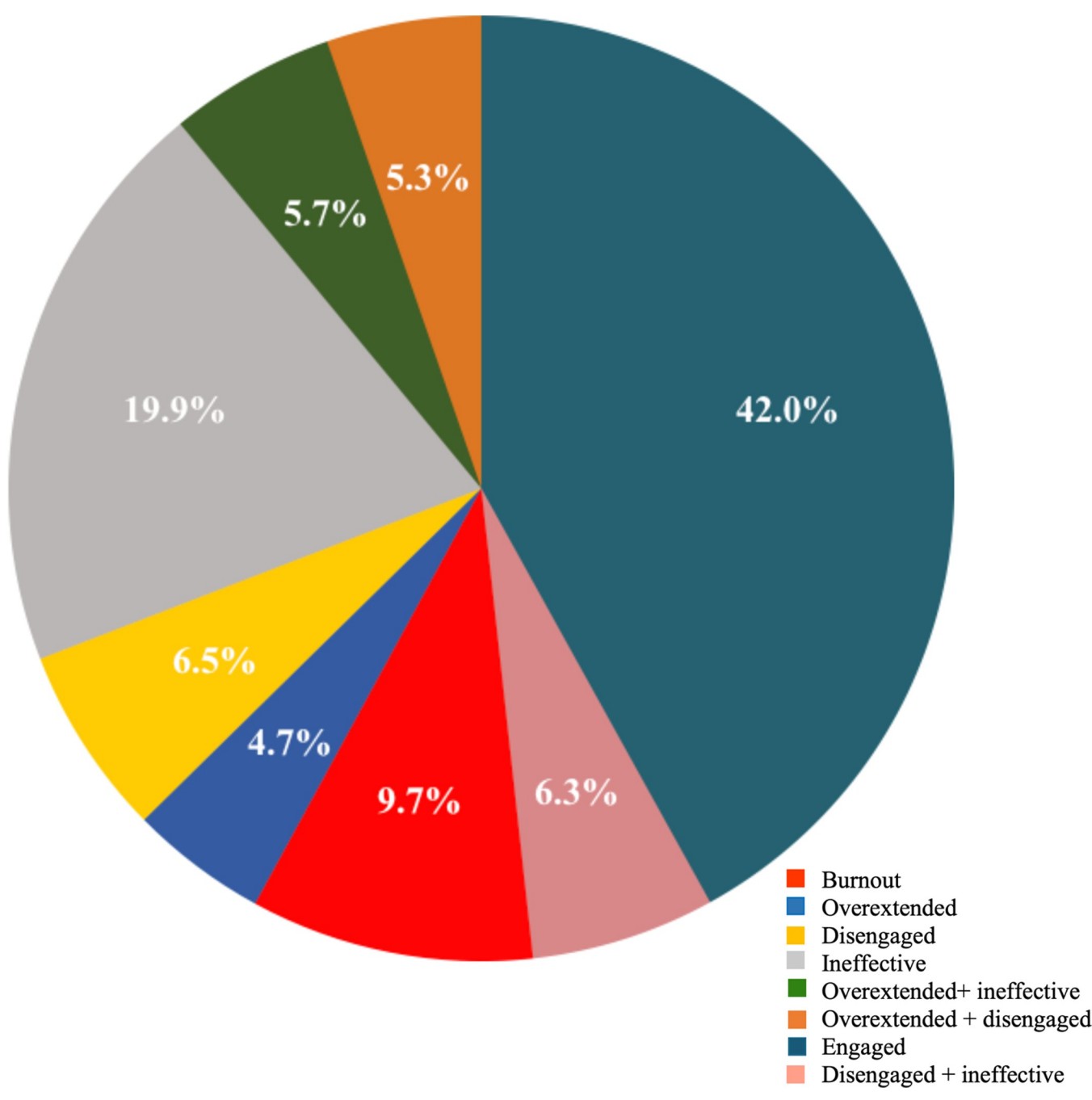

**Fig 6. Profiles of burnout scores.**

involved in the care of COVID-19 patients were more likely to experience symptoms of depression, anxiety, insomnia, and distress [27]. A Meta-Analysis revealed that nurses, women, frontline health workers and younger medical staff reported severe degrees of psychological distress, anxiety, depression and stress with low sleep quality [28].

The risk of psychological distress is higher in females (56.1%) as compared to males (49.2%) which supports the results of the previous studies [29] that reported that prevalence of anxiety and depression was more among females (39.3%) as compared to males (24.6%) among health

care professional who were involved in the COVID-19 care treatment. The regression analysis also revealed that women health workers were 55% more at risk of psychological distress as compared to men (OR = 1.55, 95% CI (1.04–2.29). Many previous studies also highlighted that the females were at higher risk, of anxiety/depression as compared to male health workers [27, 30, 31].

Among medical health care workers our study found that doctors and nurses reported higher values of GHQ as compared to other health workers (65.3% vs. 45.5%). In a study on healthcare professionals in India, 55% of medical officers suffered from moderate depression [32]. In Italy in a comparative study of psychiatric impact on general population and healthcare workers [31] younger age and female gender showed higher scores of distress and healthcare workers presented higher levels of psychiatric symptoms (anxiety and depression) than the general population. Another study on comparison of anxiety and depression scores between medical staff and administrative staff [15] using Hamilton Anxiety Scale (HAMA)and Hamilton Depression Scale (HAMD) found that levels of HAMA (4.73 vs. 3.67) and HAMD (2.41 vs. 1.86) were higher in the medical staff. These findings indicated that those health care professionals who were in close contact with COVID patients (clinical staff) revealed more psychological disorders as compared to non-clinical staff.

Our study has shown that healthcare workers who were younger (< = 34 years), single, doctors and nurses, in a higher income group (> = 20000), not residing with their family and more qualified had higher mean scores of emotional exhaustion and depersonalisation and lower scores of personal accomplishments. This is in agreement with the study in Italy [31] that found higher mean Emotional exhaustion and depersonalization score among clinical health care workers (p<0.05) as compared to non-clinical health care workers although our study found a non-significant difference in the personal accomplishment score between the two groups.

The risk of psychological distress may not simply be due to direct contact with COVID patients but rather due to the engagement with severely ill patients that require intensive care. Therefore, to minimise the risk of psychological distress and emotional exhaustion during pandemic outbreaks, hospital administrations should recruit additional staff and restrict extended workloads by giving frequent breaks to health care workers.

There are very few studies on intervention to mitigate the adverse effects of the pandemic. The World Health Organization had first released a document to recognize this risk formally [33]. According to the article, implementing the most stringent preventive measures, reducing the anxiety/stigma associated with COVID-19 transmission, and providing adequate psychological and social support will significantly lower occupational stress among the health care professionals. Zhu et al. [34] observed that implementation of psychological preventive measures and relaxation techniques in the health workers in Wuhan lowered rates of adverse psychological outcomes. Xiao et al. [17] studied the effect of social support on the mental health of 180 physicians and nurses who were treating COVID-19 infected patients in a Wuhan hospital. The researchers found that responders reported elevated levels of anxiety, stress, and self-efficacy, dependent on sleep quality and social support.

The present paper pertains to the mental health status in terms of reported burnout and psychological distress among the health care workers in 2020 due to the COVID -19 pandemic. The healthcare workers were under severe pressure to treat the surging number of cases in the hospital, with the non-availability of oxygen cylinders, medicines and the growing number of Mucormycosis cases among the patients. They are still under pressure even as India is in the third phase of its vaccination drive, with additional vaccines being given licenses for restricted use in emergency situation. Considering the above aspects, the study recommends periodic assessment of the health care workers' mental health status with need-based interventions by

the organizations. It also recommends the need to spread awareness among the health care professionals, ministries, general public on the challenges faced by health care workers that helps to improve the mental wellbeing of the healthcare workers.

## Limitation

This study did not collect any information on the past medical history of psychiatric disorders as part of this survey. Since there was no pre-COVID-19 and post-COVID-19 pandemic study conducted it was difficult to establish the extent of mental health problems and factors that were accountable for the mental health. Due to the short duration of the study and restrictions on travel, it was not possible to collect representative data from each state. The ICMR-NIOH Burnout questionnaire was an indigenous instrument that was still under validation at the time of the study.

## Conclusion

The study's findings shed light on the various mental health concerns that healthcare personnel faced during the COVID-19 pandemic, including anxiety, depression, burnout, and social stigma. Additionally, the study findings will aid in the management and planning of measures to alleviate mental health concerns among healthcare personnel in the event of future epidemic/pandemic scenarios in the country. Despite the family being the main network and care provider, with changes in the social and demographic profile, there is also a need to agree to take a new perception to resolve issues related to the medical personal involved in COVID treatment.

## Supporting information

**S1 File. Questionnaire.**
(DOCX)

**S2 File. Functional diagram to explain the flow of work.**
(PDF)

**S1 Table. Sociodemographic and job-related factors and the burnout subscale scores.**
(DOCX)

**S1 Data. Anonymized data-file in excel.**
(XLS)

## Author Contributions

**Conceptualization:** Geetha R. Menon, Sumit Aggarwal, Ravinder Singh, Tapas Chakma, Beena Thomas.

**Data curation:** Jeetendra Yadav, Tapas Chakma, Murugesan Periyasamy, Bijaya Kumar Mishra, Maribon Viray, K. H. Jitenkumar Singh, Chandra Suresh, Dhanalakshmi A., Basilea Watson, Pradeep Selvaraj, Gladston Xavier, Sairu Philip, Geethu Mathew, Alice David, Raman Swathy Vaman, Abey Sushan, Kiran Jakhar, Asha Ketharam, Ranjan Prusty, U. Venkatesh, Srikanta Kanungo, Krushna Sahoo, Swagatika Swain, Anniesha Lyngdoh, Jochanan Diengdoh, Phibawan Syiemlieh, AbuHasan Sarkar.

**Formal analysis:** Geetha R. Menon, Jeetendra Yadav.

**Funding acquisition:** Sumit Aggarwal, Ravinder Singh, Tapas Chakma.

**Investigation:** Simran Kaur, Chitra Venkateswaran, Prashant Kumar Singh, Rakesh Balachandar, Ragini Kulkarni, Ashoo Grover, Bijaya Kumar Mishra, Maribon Viray, Kangjam Rekha Devi, K. H. Jitenkumar Singh, K. B. Saha, P. V. Barde, Chandra Suresh, Basilea Watson, Gladston Xavier, Jaideep Menon, Sairu Philip, Alice David, Abey Sushan, Shalini Singh, Asha Ketharam, Ranjan Prusty, Jugal Kishore, Subrata Kumar, AbuHasan Sarkar, Gajanan Velhal, Swapnil Kharnare, Deepika Nandanwar.

**Methodology:** Geetha R. Menon, Jeetendra Yadav, Tapas Chakma, Murugesan Periyasamy, Denny John.

**Project administration:** Sumit Aggarwal.

**Resources:** Simran Kaur, Ashoo Grover, K. B. Saha, Beena Thomas.

**Software:** Jeetendra Yadav.

**Supervision:** Geetha R. Menon, Tapas Chakma, P. V. Barde, Beena Thomas, Denny John, Shalini Singh, Jugal Kishore, M. Vishnu Vardhana Rao, Samiran Panda.

**Validation:** Chitra Venkateswaran, Prashant Kumar Singh, U. Venkatesh, Samiran Panda.

**Writing – review & editing:** Geetha R. Menon, Sumit Aggarwal, Ravinder Singh, Simran Kaur, Tapas Chakma, Murugesan Periyasamy, Chitra Venkateswaran, Prashant Kumar Singh, Rakesh Balachandar, Ragini Kulkarni, Ashoo Grover, Bijaya Kumar Mishra, Maribon Viray, Kangjam Rekha Devi, K. H. Jitenkumar Singh, K. B. Saha, P. V. Barde, Beena Thomas, Denny John, Jaideep Menon, Shalini Singh, Asha Ketharam, Ranjan Prusty, Jugal Kishore, U. Venkatesh, Subrata Kumar, Srikanta Kanungo, Krushna Sahoo, AbuHasan Sarkar, M. Vishnu Vardhana Rao, Samiran Panda.

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
