## [Decision Letter · Decision Letter 0]

20 Sep 2021

PONE-D-21-26475Psychological Distress and Burnout among Healthcare Worker during COVID-19 Pandemic in India- A cross-sectional studyPLOS ONE

Dear Dr. Menon,

Thank you for submitting your manuscript to PLOS ONE. After careful consideration, we feel that it has merit but does not fully meet PLOS ONE’s publication criteria as it currently stands. Therefore, we invite you to submit a revised version of the manuscript that addresses the points raised during the review process.

We look forward to receiving your revised manuscript.

Kind regards,

M. Shamim Kaiser, PhD

Academic Editor

PLOS ONE

Journal Requirements:

2. Please provide additional details regarding participant consent. In the ethics statement in the Methods and online submission information, please ensure that you have specified whether consent was informed.

Reviewers' comments:

Reviewer's Responses to Questions

**Comments to the Author**

1. Is the manuscript technically sound, and do the data support the conclusions?

Reviewer #1: Yes

Reviewer #2: No

2. Has the statistical analysis been performed appropriately and rigorously? 

Reviewer #1: I Don't Know

Reviewer #2: No

3. Have the authors made all data underlying the findings in their manuscript fully available?

Reviewer #1: No

Reviewer #2: Yes

4. Is the manuscript presented in an intelligible fashion and written in standard English?

Reviewer #1: Yes

Reviewer #2: No

5. Review Comments to the Author

Reviewer #1: The authors conducted a cross sectional study and explored the psychological distress and burnout rate among the healthcare workers during COVID-19 pandemic. The aim of the study was to determine the burnout levels and factors associated with the risk of psychological distress among healthcare workers (HCW) engaged in the management of COVID-19 in India. This is an interesting representation of data and several significant insights were found which may help in future pandemic management. However, the authors need to address some issues before acceptance of the manuscript.

1. A table on questionnaire need to be added. Why did the authors specifically choose the GHQ-5 and ICMR-NIOH questionnaires? Why not other questionnaires ?

2. The authors did not mention the sampling procedure of the study. It should be mentioned.

3. Several statistical tests have been performed. A short description of tests may help the readers to understand the article even better.

4. There was no city wise distribution of burnout levels and psychological distress rate which need to be done. A comparison with average covid cases with the burnout and psychological distress may be interesting.

5. This manuscript lacks visual representation of data. More visual representation may be added which will help the reader to understand the data.

6. There were several typos, comma and spacing problems.

7. Results written in abstract need to be rewritten. It was not clear to the reviewer.

8. Causes of psychological distress can be shown in pictorial form which may increase the readability of the article.

9. Overall writing needs to be improved. Reviewer found difficulty in understanding the result discussion.

10. A specific section for abbreviation or terminology maybe added which will increase the readability.

Reviewer #2: This paper proposes Psychological Distress and Burnout among Healthcare Worker during COVID-19 Pandemic in India- A cross-sectional study.

In this paper the authors took 12 different cities for study purpose but didn’t give any justification of choosing those cities for research purpose.

The authors failed to mention what time of a day they have conducted the interview which may have a major impact while researching on psychological impact of healthcare workers for this study.

The authors should include a functional diagram in the paper to explain the flow of the work as well as mathematical modelling to explain the multiple logistic regression process that has been used for adjusting the risk factors.

All the figures are too blur to understand and therefore should be redraw.

There are grammatical errors in the paper which reduce the readability of the article. These errors should be corrected.

6. PLOS authors have the option to publish the peer review history of their article (what does this mean?). If published, this will include your full peer review and any attached files.

Reviewer #1: No

Reviewer #2: No

---

## [Author Response · Author response to Decision Letter 0]

26 Oct 2021

Author’s response to the Editor’s comments

Comment: A rebuttal letter that responds to each point raised by the academic editor and reviewer(s). You should upload this letter as a separate file labelled 'Response to Reviewers'.

Response: Rebuttal letter uploaded as a separate file 

Comment: A marked-up copy of your manuscript that highlights changes made to the original version. You should upload this as a separate file labelled 'Revised Manuscript with Track Changes'.

Response: A marked up copy of the manuscript “PONE-D-21-26475_marked up” uploaded

Comment: An unmarked version of your revised paper without tracked changes. You should upload this as a separate file labelled 'Manuscript'

Response: An unmarked version of the manuscript” Manuscript” uploaded

Author’s response to the Reviewer’s comments

Please find the comments and their respective responses below

Reviewer #1 comments and Author’s responses:

Comments: The authors conducted a cross sectional study and explored the psychological distress and burnout rate among the healthcare workers during COVID-19 pandemic. The aim of the study was to determine the burnout levels and factors associated with the risk of psychological distress among healthcare workers (HCW) engaged in the management of COVID-19 in India. This is an interesting representation of data and several significant insights were found which may help in future pandemic management. However, the authors need to address some issues before acceptance of the manuscript.

Response: We wish to thank the reviewer for motivating feedback. The authors are grateful to the esteemed reviewer for appreciating the relevance of our work. We have considered all the suggestions given by reviewer# 1 in the revised version of the manuscript.

Our detailed reply to each and every point raised by the Reviewer #1 is given below:

Query 1: A table on questionnaire need to be added. Why did the authors specifically choose the GHQ-5 and ICMR-NIOH questionnaires? Why not other questionnaires?

Response: Thanks to the reviewer for this important observation. The questionnaire has been attached as a supporting document (S1). The reason for using GHQ-5 and ICMR-NIOH is given below and the same has been incorporated in the revised version of the manuscript.

The GHQ-5, is a shorter screening tool (not a diagnostic tool) which has 5 questions with better discriminators for psychological distress derived from GHQ-12 that involves royalty for usage. GHQ-5 is validated on the Indian population with a sensitivity of 86%, specificity of 95.8% and misclassification rate of 8.3% (Reference). For this study that was done telephonically and needed quick assessment of the psychological state of the respondent with minimum number of questions, GHQ-5 was found to be an effective tool with regard to time and process. It is also available for use in public domain as compared to GHQ-12. 

Reference: C. Shamasunder, T. G. Sriram, S. G. Murali Raj, and V. Shanmugham, “Validity of a short 5-item version of the general health questionnaire (g.h.q).,” Indian J. Psychiatry, vol. 28, no. 3, pp. 217–9, Jul. 1986.

ICMR-NIOH burnout questionnaire is a shortened and easier version of the Maslach Burnout inventory (MBI) (Reference 1). The questionnaire is a 19 item tool that covers three dimensions of burnout: a) emotional exhaustion (EE, 11 items); (b) depersonalisation (DP, 5 items) and (c) personal accomplishment (PA, 3 items) on a 3-point scale starting from 1,” never,” 2, “sometimes” and 3, “always”. Higher scores on the EE and DP subscales and lower scores on the PA subscale indicate high degrees of experienced burnout. It was developed by the scientists of the ICMR-National Institute of Occupational Health (NIOH) and was under validation for Indian settings during the study. Now it has been validated and published (Reference 2). It was freely available for use whereas the MBI inventory is copyrighted and cannot be reproduced without permission from the developers. Due to the ongoing pandemic and need to make a quick assessment of burnout among healthcare workers, we used the ICMR-NIOH questionnaire. 

Reference 1: C. Maslach, S. E. Jackson, and M. P. Leiter, “Maslach Burnout Inventory: Third edition.,” in Evaluating stress: A book of resources., Lanham, MD, US: Scarecrow Education, 1997, pp. 191–218.

Reference 2. S. Balachandar, R., Ketharam, A., & Bharath, “Development and validation of tool for screening occupational mental health and workplace factors influencing it,” PsyArXiv. July 1., pp. 1–17, 2021.

Query 2: The authors did not mention the sampling procedure of the study. It should be mentioned.

Response: Thanks for your observation. The detailed sampling procedure and selection of participants has been described in the revised version of the manuscript.

Query 3: Several statistical tests have been performed. A short description of tests may help the readers to understand the article even better.

Response: Thanks for your observation. The step by step statistical procedures/tests with equation have been added in the statistical analysis section of the revised manuscript.

Query 4: There was no city wise distribution of burnout levels and psychological distress rate which need to be done. A comparison with average COVID cases with the burnout and psychological distress may be interesting.

Response: The purpose of the study was to gain a quick insight on the psychological state and burnout levels of healthcare workers in India when state wise COVID infection rates were varying. A representative sample in each state was difficult to obtain due to the government restrictions on travel and shorter duration of the study. A minimum sample size of 904 was determined to obtain the distribution at the national level and not at the city/state level. Hence it would be inappropriate to provide the city wise data in the manuscript. 

We appreciate the reviewers comment to compare the COVID cases with the burnout levels and psychological distress. However, we could not do this since the data on number of COVID cases was not verified from official records due to the inability of the investigators to physically approach the facility because of ongoing pandemic and lockdown. 

Query 5: This manuscript lacks visual representation of data. More visual representation may be added which will help the reader to understand the data.

Response: We thank the reviewer for the observation. We have displayed the results in graphics (Fig1-6) for better visual representation in the revised manuscript. 

Query 6: There were several typos, comma and spacing problems.

Response: We appreciate the observation. We have corrected for grammatical errors in the revised version of the paper. 

Query 7: Results written in abstract need to be rewritten. It was not clear to the reviewer.

Response: We have rewritten the results in the abstract. 

Query 8: Causes of psychological distress can be shown in pictorial form which may increase the readability of the article.

Response: We appreciate the reviewer for this observation. The factors associated with risk of psychological distress has been shown in pictorial form in the revised manuscript Fig3-4.

Query 9: Overall writing needs to be improved. Reviewer found difficulty in understanding the result discussion.

Response: We appreciate the observation. Overall writing has been improved to bring more clarity in the revised manuscript. 

Query 10: A specific section for abbreviation or terminology maybe added which will increase the readability.

Response: We have added a section on list of acronyms. 

Reviewer’s #2 comments and Authors’ responses:

Comment: Reviewer #2: This paper proposes Psychological Distress and Burnout among Healthcare Worker during COVID-19 Pandemic in India- A cross-sectional study.

Response: The authors are grateful to the esteemed reviewer for appreciating the relevance of our work. We have considered all the suggestions given by reviewer# 2 in the revised version.

Our detailed reply to each and every point raised by the Reviewer’s #2 is given below:

Comment 1: In this paper the authors took 12 different cities for study purpose but didn’t give any justification of choosing those cities for research purpose.

Response: The study was an ICMR multicentric Task Force project. The sites were chosen based on the response of the site investigators to a call for proposals by the ICMR National Task Force on Operation Research on COVID-19. We selected those implementing agencies who submitted the concept proposal with similar objectives. These agencies were located in 10 states that broadly represented different regions of the country. Due to the lockdown rules and travel regulations in different cities, from each state one city was selected where the implementing agency was located except in Kerala and Odisha where two cities were selected as per the convenience of site investigators.

Comment 2: The authors failed to mention what time of a day they have conducted the interview which may have a major impact while researching on psychological impact of healthcare workers for this study.

Response: Thanks for your observation. As mentioned in the revised manuscript the investigators fixed an appointment with the healthcare workers as per their convenience so as to ensure that their duty time or leisure time was not disturbed. 

Comment 3: The authors should include a functional diagram in the paper to explain the flow of the work as well as mathematical modelling to explain the multiple logistic regression process that has been used for adjusting the risk factors.

Response: We have added the functional diagram (S2) and have described the mathematical modelling procedures in the statistical analysis section of the revised manuscript. 

Comment 4: All the figures are too blur to understand and therefore should be redraw.

Response: We appreciate the observation of the reviewer and all the figures have been redrawn Fig1-6. 

Comment 5: There are grammatical errors in the paper which reduce the readability of the article. These errors should be corrected.

Response: We appreciate the observation. We have corrected for grammatical errors in the revised version of the paper.

---

## [Decision Letter · Decision Letter 1]

31 Jan 2022

PONE-D-21-26475R1Psychological Distress and Burnout among Healthcare Worker during COVID-19 Pandemic in India- A cross-sectional studyPLOS ONE

Dear Dr. Menon,

Thank you for submitting your manuscript to PLOS ONE. After careful consideration, we feel that it has merit but does not fully meet PLOS ONE’s publication criteria as it currently stands. Therefore, we invite you to submit a revised version of the manuscript that addresses the points raised during the review process.

We look forward to receiving your revised manuscript.

Kind regards,

M. Shamim Kaiser, PhD

Academic Editor

PLOS ONE

Journal Requirements:

Reviewers' comments:

Reviewer's Responses to Questions

**Comments to the Author**

1. If the authors have adequately addressed your comments raised in a previous round of review and you feel that this manuscript is now acceptable for publication, you may indicate that here to bypass the “Comments to the Author” section, enter your conflict of interest statement in the “Confidential to Editor” section, and submit your "Accept" recommendation.

Reviewer #1: All comments have been addressed

Reviewer #2: All comments have been addressed

2. Is the manuscript technically sound, and do the data support the conclusions?

Reviewer #1: Yes

Reviewer #2: Yes

3. Has the statistical analysis been performed appropriately and rigorously? 

Reviewer #1: Yes

Reviewer #2: Yes

4. Have the authors made all data underlying the findings in their manuscript fully available?

Reviewer #1: No

Reviewer #2: Yes

5. Is the manuscript presented in an intelligible fashion and written in standard English?

Reviewer #1: Yes

Reviewer #2: Yes

6. Review Comments to the Author

Reviewer #1: I am satisfied with how the authors addressed my comments. With some minor revisions, I accept the manuscript. Make sure that every figure is clear and well-illustrated. Captions should be self-explanatory. Particularly, Figures 3, 4, and 5 are blurry and difficult to read.

Reviewer #2: (No Response)

7. PLOS authors have the option to publish the peer review history of their article (what does this mean?). If published, this will include your full peer review and any attached files.

Reviewer #1: **Yes: **Md Jaber Al Nahian

Reviewer #2: **Yes: **Dr. Risala Tasin Khan

---

## [Author Response · Author response to Decision Letter 1]

4 Feb 2022

Author’s response to the Reviewer’s comments

Please find the comments and their respective responses below

Comments to the Author

1. If the authors have adequately addressed your comments raised in a previous round of review and you feel that this manuscript is now acceptable for publication, you may indicate that here to bypass the “Comments to the Author” section, enter your conflict of interest statement in the “Confidential to Editor” section, and submit your "Accept" recommendation.

Reviewer #1: All comments have been addressed

Reviewer #2: All comments have been addressed

Response: Thanks for the reviewer’s appreciation

2. Is the manuscript technically sound, and do the data support the conclusions?

Reviewer #1: Yes

Reviewer #2: Yes

Response: Thanks for the reviewer’s appreciation

3. Has the statistical analysis been performed appropriately and rigorously? 

Reviewer #1: Yes

Reviewer #2: Yes

Response: Thanks for the reviewer’s appreciation

4. Have the authors made all data underlying the findings in their manuscript fully available?

Reviewer #1: No

Response: Thanks for the observation. We have now uploaded the data as per the PLOS DATA POLICY

Reviewer #2: Yes

Response: Thanks for the valuable comments 

5. Is the manuscript presented in an intelligible fashion and written in standard English?

Reviewer #1: Yes

Reviewer #2: Yes

Response: Thanks for the reviewer’s appreciation

6. Review Comments to the Author

Reviewer #1: I am satisfied with how the authors addressed my comments. With some minor revisions, I accept the manuscript. Make sure that every figure is clear and well-illustrated. Captions should be self-explanatory. Particularly, Figures 3, 4, and 5 are blurry and difficult to read.

Response: Thanks for the accepting the manuscript. The authors appreciate the reviewers for their valuable inputs. We have now uploaded then revised figures. 

Reviewer #2: (No Response)

Response: Thanks to the reviewer.

---

## [Editor Report · Decision Letter 2]

22 Feb 2022

Psychological Distress and Burnout among Healthcare Worker during COVID-19 Pandemic in India- A cross-sectional study

PONE-D-21-26475R2

Dear Dr. Menon,

We’re pleased to inform you that your manuscript has been judged scientifically suitable for publication and will be formally accepted for publication once it meets all outstanding technical requirements.

Kind regards,

M. Shamim Kaiser, PhD

Academic Editor

PLOS ONE

Additional Editor Comments (optional):

NA

Reviewers' comments:

NA

---

## [Editor Report · Acceptance letter]

28 Feb 2022

PONE-D-21-26475R2 

Psychological Distress and Burnout among Healthcare Worker during COVID-19 Pandemic in India- A cross-sectional study 

Dear Dr. Menon:

I'm pleased to inform you that your manuscript has been deemed suitable for publication in PLOS ONE. Congratulations! Your manuscript is now with our production department. 

Kind regards, 

on behalf of

Dr. M. Shamim Kaiser 

Academic Editor

PLOS ONE